# Exploring the Effects of Guided vs. Unguided Art Therapy Methods

**DOI:** 10.3390/bs10030065

**Published:** 2020-03-07

**Authors:** Ana Maria Costa, Rui Alves, São Luís Castro, Selene Vicente, Susana Silva

**Affiliations:** Center for Psychology, Faculty of Psychology and Education Sciences, University of Porto, 4200-135 Porto, Portugal; anamariacosta23@gmail.com (A.M.C.); ralves@fpce.up.pt (R.A.); slcastro@fpce.up.pt (S.L.C.); svicente@fpce.up.pt (S.V.)

**Keywords:** art, program evaluation, wellness, neuropsychology

## Abstract

Art therapy has become known by its psychosocial and affective impact, but not so much by its effects on cognitive functioning. Based on a comparison between art therapy and music-making programs, we hypothesized that guided methods—dominant in music-making programs and characterized by an emphasis on execution (play the piece, produce the visual object) rather than ideation (conceive the visual object)—could boost the cognitive effects of art-making. We also hypothesized that removing ideation from the process with guided methods could decrease psychosocial/affective benefits. In order to test our hypotheses, we compared the effects of two art therapy methods on cognitive vs. psychosocial/affective domains. We implemented a short-term longitudinal study with patients with schizophrenia showing both psychosocial/affective and cognitive deficits. The sample was divided into two groups: unguided, instructed to ideate art pieces and execute them without external guidance, vs. guided, instructed to execute predefined art pieces following externally provided guidelines. There was no evidence that guided methods boost cognitive effects, since these were equivalent across groups. However, psychosocial/affective benefits were enhanced by unguided methods, suggesting that therapeutic methods can make a difference. Our study contributes to raising important new questions concerning the therapeutic mechanisms of art therapy.

## 1. Introduction

Art therapy refers to the use of artistic practices with therapeutic purposes [1,2]. The designation became synonymous with the practice of *visual arts* such as drawing, painting, sculpture and collage [3], thus not including other art-based therapies such as music or dance therapy. As any visual art, art therapy includes a combination of imagery-related-ideation (conceiving the visual object, *ideation* hereafter) with *execution* (using tools and materials to produce the art piece). The extent to which the art therapist constrains the patient’s activity at these two levels contributes to define the type of *therapeutic method* that is used. For instance, a classic dichotomy in art therapy methods contrasts directive methods—where the topic for ideation (e.g., patient’s fears) is predefined by the therapist—with non-directive ones, where the patient is free to choose his/her own ideation topic [4,5,6].

In the present study, we compare non-directive methods—where patients are responsible for both ideation and execution, with few or no guidelines given—with an extreme version of directive methods, where ideation is totally absent, and patients are provided with instructions on how to use tools and materials so as to produce a predefined shape or image. In order to distinguish our manipulation from the classic directive vs. non-directive dichotomy, we referred to our two methods as *guided* (extreme version of directive) and *unguided*, respectively. Our main goal was to determine whether unguided methods could boost cognitive effects from art therapy. Please note that we do not assume that art making is necessarily a creative process. According to Schoop’s definition [7], creation is an autonomous process of shaping in which all decisions are bound to the creating person, and this is not always so in artistic practice: for instance, a musician who plays a piece by following all indications from the score is not deciding anything, and the same goes for someone who copies a drawing—provided that s/he does not add anything to the drawing intentionally. Therefore, when comparing unguided with guided methods, we were contrasting creative practices (unguided method) with not-necessarily creative ones (guided).

Since its beginnings, art therapy remained focused on *psychosocial*—e.g., improving well-being, quality of life and communication, as well as *affective* goals—e.g., reducing anxiety levels and enhancing positive mood [8]. Adrian Hill (1895–1977), the creator of the ‘art therapy’ designation, developed his first therapeutic experiments when trying to fight the affective and psychosocial consequences of tuberculosis [9]. Since then, empirical studies (see Table 1) have prioritized psychosocial and affective areas across various clinical and non-clinical groups [10]: In psychotic patients, case reports [11,12,13] and quantitative studies with control groups [14,15,16,17,18] reported decreased anxiety [11,16,17] and depression [17], reduction in negative symptoms such as apathy or anhedonia [14,17], as well as increased self-esteem [12,14,15], sense of self [12,13], socialization and compliance with rules and programs [17,18] following art therapy. In non-psychotic psychiatric patients, systematic reviewing has identified a positive impact of art therapy on depression, anxiety and quality of life in depressed, anxious and phobic patients [19], as well as on depression and trauma in traumatized adults [20]. Outside psychiatry, the effects of art therapy on well-being after cancer diagnosis [21], on anger and conflict management in offenders [22,23], as well as on stress prevention in healthy or at-risk individuals [24] are all well-established [10].

In contrast, evidence for an impact of art therapy on the *cognitive* domain remains poor. Some studies with older adults—where concerns with cognitive preservation could be expected—did not even measure cognitive outcomes (see Table 1), focusing instead on emotion, self-esteem, anxiety [25], well-being [26] or depression [27]. Among the few available studies targeting the cognitive outcomes of art therapy on dementia, some point to enhancements of attention and memory [28,29]. Nevertheless, a recent systematic review concluded that art therapy may be useful for addressing the emotional issues of dementia, but not cognitive decline per se [30]. In an even more recent review [31], the cognitive outcomes of art therapy in people with dementia were classified as ‘very low’. In a time when cognitive rehabilitation is a major societal challenge due to an ageing population, it is crucial to know whether and how art therapy is a suitable means to address such challenge.

One reason why art therapy has not been related to cognitive benefits may be the dominant use of unguided therapeutic methods. Guided approaches—as defined here—lie outside the tradition of art therapy, in which expression has been central [32]. For instance, Nise da Silveira (1905–1999)—who had a prominent role in the history of art therapy—drew upon Jungian concepts to implement therapeutic programs in patients with schizophrenia [33], and she did it by using totally unguided methods. According to Nise da Silveira, the unguided ideation of the schizophrenic patient would bring her/his internal world closer to consciousness, allowing her/him to reorganize it and rebuild a relationship with reality. After Nise da Silveira, no great shift from this expressive approach to art therapy seems to have occurred: even McNeilly’s [4] definition of directive methods does not go beyond the possibility of predefining the theme for ideation, and the patient is still responsible for the particular visual object that is created. 

Why should guided methods favor cognitive outcomes? Unlike unguided methods, guided ones imply permanent attention and compliance with externally provided guidelines. Therefore, it is likely that guided methods maximize alertness, memory, attention and executive functioning in contrast to unguided, where patients are left alone with their own mental paths. Although we can also imagine arguments in favor of the reverse hypothesis—i.e., that unguided methods allow patients to be more autonomous and, hence, more cognitively engaged—the truth is that unguided patients will always have the choice not to be engaged: since they do not have to respond to external demands, they are free to do something random. The idea that compliance with external directives is key to cognitive boosting subtends modern (cognitive) music therapy [34], where, for instance, synchronization with an external rhythm drives cognitive-motor rehabilitation processes. It also applies to music training studies [35], which have shown cognitive effects from training and are generally grounded on instrumental execution and auditory discrimination rather than creation or expression. Thus, while art therapy seems to be focused on creation and expression using unguided methods, modern music therapy and music-based educational programs seem to value execution as implemented with guided methods. While art therapy became notable for its psychosocial and affective impact, music-based interventions stood out for their effects on cognition. Here, we hypothesize that these two aspects may be linked, and the use of guided methods—whatever the domain—may boost cognition.

Once we decide to reduce the role of ideation in the artistic process by implementing guided methods—in which patients are not responsible for conceiving the visual object—a new question emerges: do psychosocial and affective benefits remain? One argument in favor of ‘no’ is that these types of benefits seem to be strongly linked to the ideation component of art therapy, more than to that of execution. Nise da Silveira’s psychoanalytic approach to art therapy viewed psychosocial/affective improvements as a result of patients’ getting closer to their unconscious universe as they created visual shapes charged with symbolism—i.e., as they ideated. Therefore, without ideation, emotional healing would be blocked. Alternative, non-psychoanalytic accounts of the role of ideation include Koch’s Model of Embodied Aesthetics [36] and related empirical evidence that the experience of creativity (ideation)—together with empowerment, impact and freedom—mediates post-therapy improvements in well-being and self-efficacy [37]. The literature on *flow states* in art making [38,39] also highlighted the potential of ideation to increase well-being and positive mood [40], focusing on the intermediary role of flow. Flow states [41,42] refer to moments of intense concentration, experienced when someone engages in an intrinsically rewarding activity. The phenomenological experience of flow includes effortless attention, lack of worry, distorted sense of time and reduced sense of self [38]. Critically, flow states seem to arise from feelings of accomplishment and autonomy [43], and the probability of these feelings increases as one creates something of his/her own (ideates). Therefore, while replacing the patient’s ideation by externally provided directives (guided method) may increase their cognitive gains, it also seems possible that psychosocial and affective benefits decrease as we do this. An exception to the beneficial effects of ideation on the psychosocial and affective domains seems to occur when patients’ freedom to ideate leads them to focus on their own negative experiences or current problems as themes: in these cases, psychosocial and affective changes may be null or even negative [44]. This not being the case, it seems that we may expect unguided methods to win over guided ones concerning effects on well-being and mood.

In the present study, our goal was to determine if it is possible to maximize the cognitive outcomes of art therapy using guided methods and, in case this is possible, whether we lose or not the well-known psychosocial and affective impact of art therapy [8,9,10,11,12,13,14,15,16,17,18,19,20,21,22,23,24,25,26,27]. We tested the hypotheses that (1) guided methods favor cognitive benefits, while (2) unguided therapeutic methods privilege affective and psychosocial ones, implying that the latter will decrease under guided methods. 

In order to test our hypotheses, we focused on a population in which the two types of deficits—psychosocial/affective and cognitive—are clearly present, as it is the case in individuals with schizophrenia. Schizophrenia is a disorder whose diagnostic is based on evidence of inadequate social behavior as well as the inability to distinguish what is real from what is not [45]. Individuals with schizophrenia may show affective changes such as anxiety, depression, anger, loss of energy and initiative, loss of interest, as well as psychosocial deficits that include isolation, personal suffering, inadequate behavior, negligence with personal appearance and hygiene, disturbance in sleep patterns, or lack of interest in diet [46,47,48]. Cognitive deficits are also at the core of schizophrenia [49], and seem to be present early on, even at the time of the first episode of the disorder [50]. The most affected cognitive functions in schizophrenia are processing speed, working memory, verbal memory and learning, visual memory and learning, reasoning and problem solving, and attention/vigilance [51], indicating the importance of executive functioning deficits [52,53] in this disorder. Deficits in motor dexterity and spatial skills have also been pointed out [54]. Along with these cognitive changes, emotional recognition and social perception are also usually affected [55]. 

In line with our hypotheses (see above), we evaluated the psychosocial/affective and cognitive functioning of patients with schizophrenia before (pre-test) and after (post-test) a three-month intervention program, using standardized tests. Patients were divided into two groups: one group was exposed to an unguided method of art therapy, the other to a guided one. In the guided-method group, patients were provided with images to be recreated and/or copied, as well as with guidelines for manual execution. In contrast, unguided-group patients were presented images for inspiration for a very short period, after which they were invited to do whatever they wished. Apart from these differences, activities were designed to be equivalent—with the same duration, same available media and techniques (collage, painting or drawing) and, most of the times, with the same images for copy/recreation (guided) vs. inspiration (unguided). The difference between pre-test and post-test psychosocial/affective and cognitive functioning as measured by standardized tests (objective intervention-related improvement) was calculated for both groups. We expected larger cognitive improvements in the guided-method group due to the externally provided structure, and larger psychosocial/affective improvements in the unguided-method group compared to the guided-method group due to the presence of ideation.

## 2. Materials and Methods

### 2.1. Participants

Sixteen participants attending a day care center volunteered to take part in this study. They had all been diagnosed with schizophrenia after psychiatric examination All were receiving pharmacological treatment, and medical records indicated no comorbidities. Most of them lived with relatives, and some lived alone. They attended the day care center 5 days per week, from 10am to 4 pm. 

The 16-participant sample (12 male; age, *M* ± *SD*: 45.9 years ± 6.3; illness duration: 26 ± 7.2; schooling: 11.3 ± 1.1; 12 with painting experience) was split into two, and eight participants were assigned to each group—guided vs. unguided. Participants from the two groups were matched for age, schooling and illness duration. They were also matched for previous painting experience, since cross-group differences in this regard could entail different technical skills, expectations and attitudes, generating differences in therapeutic outcomes unrelated to our manipulation of therapeutic methods. Moreover, being familiar with patients’ previous experience would allow to optimize the workshop sessions. Among the 8 participants from the unguided group, two did not arrive on time from vacations to start the program, and one missed the pre-test. Thus, only 5 participants in the unguided group attended the program. From these, 2 missed the post-test, leaving us with the results from 3 participants in the unguided group, against 8 in the guided one (Table 2). There was one male participant in the unguided group and 6 in the guided one. Current pharmacological treatment included antipsychotic medication (all participants), benzodiazepines (3 in the unguided group, 4 in the guided one), anticholinergics (3 in unguided, 4 in guided), antiepileptics (2 unguided, 1 guided) and antidepressants (1 unguided, 1 guided). The average age of illness onset was low in both groups (Table 1): this suggests short durations of untreated psychosis, since it is likely that initial awareness of the disease was followed by treatment. Illness duration was high in both groups. One participant in the guided group and 2 in the unguided had no artistic experience at all; the others (2 vs. 6) had some. The initial group matching was overall preserved (Table 2), except for gender. 

Ethics approval was provided by the day care center (Project identification code 2017/01). All participants signed informed consent in accordance with the Declaration of Helsinki.

### 2.2. Procedure

The pre-test, post-test and art therapy sessions took place at the day care center attended by participants. At the beginning of the pre-test session (one week before the intervention program), participants signed informed consent and filled in a brief sociodemographic and clinical questionnaire addressing issues such as schooling, previous painting experience, illness duration or medication. Then they took part in a 30-minute evaluation protocol with standardized tests for assessing affective/psychosocial vs. cognitive functioning (see Materials, below). The protocol was performed by a neuropsychologist who was completely unaware of the assignment of each participant to a specific group. 

Each participant took part in a 90-minute art therapy session twice a week (see Intervention program, below). Both groups had sessions on the same day. The program lasted for three months (24 sessions per group). The first author of this paper (art therapist) was responsible for the art therapy sessions.

One week after the last therapy session, patients underwent a 30-minute post-test evaluation protocol, based on the same standardized tests as the pre-test (see Materials). They also filled in a self-report control questionnaire whose main goal was to assess their own perception of personal change during the previous three-month period (subjective intervention-related change). 

### 2.3. Materials

The initial questionnaire addressed sociodemographic and clinical information, leisure time activities and painting experience.

Patients’ pre- and post-intervention assessment included measures of global cognition, executive functioning (cognitive domain), anxiety and depression, well-being and quality of life (affective/psychosocial). The following tools were used for the cognitive domain:

MoCA (Montreal Cognitive Assessment [56], Portuguese version by [57]): MoCA is a brief screening tool for global cognition, specifically developed to assess mild cognitive impairment. It is organized into eight cognitive domains: (1) Visuo-spatial and executive domain; (2) naming; (3) memory; (4) attention; (5) language; (6) abstraction; (7) delayed recall; and (8) orientation. Maximum score is 30 points.

Verbal Phonemic Fluency: This test measures verbal generation. Participants are asked to say aloud for one minute as many words as they can, beginning with letters M, R and P. 

Clock-Drawing test (Portuguese adaptation by [58]: This is a quick test that allows to evaluate a complex set of visuo-constructive and visuo-spatial competences, executive functioning and knowledge of numbers. The subject must draw a clock, write down the numbers, and draw the pointers to indicate a certain time, usually 11.10 am. 

TMT-A and B (Trail Making Test, A and B; Portuguese adaptation by [59]): TMT consists of two tasks, A and B. In task A, the goal is to connect a set of numbers in numerical order (from 1 to 25) in the shortest time possible. In task B, the objective is to alternately link numbers (1 to 13) and letters (A through L), in numerical and alphabetical order (e.g., 1–A, 2–B, 3–C) as quickly as possible. The test assesses sustained and divided attention, processing speed, cognitive flexibility, coordination, visual exploration and sequencing. Scores were computed according to the norms of [56].

Stroop test (words and colours test; Portuguese version by [60]): The Stroop test evaluates executive processes such as cognitive flexibility, selective attention, resistance to interference and inhibition of responses. It consists of three tasks: (1) reading words, in which the subject must read 100 colour names (green, red and blue); (2) naming the colour, in which the subject must name the print colour of 100 sequences of three XXX (xxx, xxx, xxx); (3) naming the colour of 100 words, in which the subject should not read the word but rather name the colour in which it is printed. Each task has a time limit of 45 seconds and the total number of correct responses is counted. In addition to the scores obtained in each task, the score for the interference effect was also calculated.

Spatial localization test from the Weschler Memory Scale-III; (WMS-III; [61]; Portuguese version of [59]): The spatial localization test is subdivided into two tasks: in the first, the examiner taps on a sequence of cubes arranged in a three-dimensional tray, and the task of the subject is to touch the cubes in the same order. In the second task, the subject must repeat sequences performed by the examiner in reverse order. The task requires the subject’s ability to maintain a visuospatial sequence of events in memory while planning the correct response. Correct answers were counted separately for direct and inverse order, and then summed and converted into a scalar score.

In order to assess the affective/psychosocial domains, we used the following tests:

HADS (Hospital Anxiety and Depression; Portuguese version of [62]): HADS is a self-report scale that takes about 10 minutes to complete. It consists of 14 items, divided into two subscales assessing levels of anxiety and depression respectively. Values between 0 and 7 are considered normal; between 8 and 10 they indicate light levels of anxiety or depression; between 11 and 14 the levels are moderate, and between 15 and 21 they are severe.

World Health Organization Quality of Life questionnaire (WHOQOL-Bref; Portuguese version of [63]): This self-report questionnaire on quality of life includes four domains: physical, psychological, social relations and environment. It consists of 26 items, two of which relate to general perception of quality of life (G1) and general perception of health (G2), while the remaining 24 characterize the four domains (6 items per domain). Each item is presented in the form of a 5-point Likert scale. Higher scores indicate an increased quality of life. For each domain, the average of the responses was calculated. Please note that the WHOQOL-Bref is not recommended for static assessment of schizophrenia symptoms—i.e., to qualify the patient’s level of quality of life—since there seems to be a tendency for schizophrenic patients to report a high degree of satisfaction even under adverse life conditions [64]. However, since our goal was to assess intervention-related changes (relative differences, between post- and pre-test) rather than provide a static diagnosis, we assumed that this known bias would not matter, that is, if participants overrated their quality of life, they would do it during both the pre-test and the post-test.

The self-report control questionnaire addressed patients’ perception of changes that might have co-occurred with the intervention program. They were asked whether they felt any improvement or deterioration concerning (a) emotional balance and well-being, (b) openness to communication (affective/psychosocial domain), (c) capacity for executing mental tasks (cognitive domain) and (d) performance in daily activities (affective/psychosocial and cognitive domains). In order to tap into the consequences of ideation, we asked them about the meaning of their art pieces. They were also asked whether they had had any disturbing life event, and if they had initiated other activities during the time of the intervention program, to rule out possible influences outside the intervention program. Please note that we did not ask for any details concerning disturbing life events, to avoid triggering negative feelings. We also made sure that the day care center psychologists were present and vigilant when participants filled in the questionnaires, in case any negative reaction would emerge. There were no signs of disturbance in any patient.

### 2.4. Intervention Program

Participants were gathered in small groups for the art therapy sessions, but each of them developed his/her individual work. We designed five different activities, to grant diversity and keep participants engaged. Each session focused on a single activity, which could span several sessions. Each activity was based on a specific technique, common to both groups. The groups differed in (a) the presence of ideation (yes/unguided group vs. no/guided group) and (b) the provision of guidelines for execution by the therapist (no/unguided vs. yes/guided). Guided activities (guided group) were designed to train specific executive functions such as focused attention (all activities) planning, strategy definition, organization (collage, dry pastel, painting on canvas, watercolor), cognitive flexibility and inhibition (watercolor and drawing). The unguided versions of these activities were then derived, by keeping all things equal (techniques, stimuli) but the method (see Figure 1 for an example). Given that the guided approach most often implied an initial moment of stimulation (observing paintings for recreation, focusing attention on aspects of the environment) and these initial moments could be enough to generate some form of aesthetic experience, we decided to introduce an analogue component of initial stimulation in the unguided group. Therefore, whenever possible, we stimulated unguided participants with the same paintings as the guided group, but we presented these merely as means for getting inspiration, in case unguided participants got blocked for ideas. It is important to remark that drawing upon these inspirational stimuli was not mandatory, and unguided participants were completely free to ignore them when creating their work. In some cases, we considered that presenting the same paintings as in the guided group could have negative consequences (e.g., as with figurative paintings, see dry pastel activity, below). In these cases, we violated the principle of maximum cross-group equivalence and stimulated unguided participants with poems instead. We next describe the two variants (guided vs. unguided) of each technique-based activity. 

Collage activity: Guided-group participants were invited to recreate paintings by Kandinsky and Miró. We chose these works because of their characteristic geometric shapes, which facilitate the implementation of collage techniques. After carefully observing the image, they had to draw the geometric shapes seen in the painting on glossy paper, using the right color; for example, if the painting had a red triangle, they would have to trim the triangle on the red gloss sheet. Shapes should then be cut and positioned on a cardboard according to the image. Unguided-group participants were given glue, paper and cardboard, and were then asked to make a collage. They were shown the same paintings as the guided group for about one minute for inspiration, and then they were left free to create.

Dry pastel/oil pastel painting activities: participants in the guided group were instructed to recreate paintings by Claude Monet using dry pastel, and paintings by Van Gogh using oil pastel. These works were used as targets because of their relatedness to the dry pastel and oil pastel techniques. In both cases, participants had to draw the painting first, and then fill it with color so as to get as close as possible to the originals. The dry pastel allows for greater ease in color mixing due to its softness, and the oil pastel is a material that allows to create texture. Participants in the unguided group were given the same materials, and they were left free to create. Unlike the collage activity (as well as painting-on-canvas, see below), the materials we showed to the unguided group for inspiration were different from the ones we showed the guided group for replication. The reason was that, contrary to those of Kandinsky, Miró (collage) and Amadeo de Sousa Cardoso (painting-on-canvas), Claude Monet’s paintings were figurative. Observing figurative paintings for inspiration could either constrain unguided participants ideation (they could be tempted to depict the same type of object), or generate too much strain (participants feeling challenged to figurate real-world objects with technical accuracy without guidance), and we did not want to risk that. Therefore, participants in the unguided group were given poems for inspiration. 

Painting-on-canvas activities: participants in the guided group had to recreate a painting by Amadeo de Sousa Cardoso using brushes and acrylic paints. Again, the technical characteristics of the painting (texture and color) dictated our choice to use it as reference for recreation. They were asked to make three overlapping layers to focus either on color, shape or detail. Unguided-group participants were shown the same stimulus image for three minutes for inspiration, and then were set free to paint their own canvas.

Watercolor activities: the guided-group participants were first asked to fill a sheet of paper by creating spots and splashes with a brush; later, they were asked to choose a small spot and to represent it as faithfully as possible on a larger scale. Unguided-group participants were given the same material and asked to use their creativity to explore color. 

Drawing activities: participants in the guided group did a series of drawing exercises. These included (a) void design, in which they were told to reproduce the shape of the background, instead of the shape of the figure; (b) reversed drawing, where they should copy a scene while not following the natural sequence of objects; (c) blind drawing, where they drew several objects without looking at the paper. Participants in the unguided group were left free to draw any image on the paper. They were given excerpts of poems for inspiration, and they were also invited to use their own positive memories.

### 2.5. Data Analyisis

For all standardized tests (see Materials) except HADS (anxiety and depression evaluation, 0–21 scale in each dimension) and WHOQOL-BREF (evaluation of well-being in various domains, scale from 0 to 100 for each domain), z-scores were considered. In most instruments, higher scores indicate superior performance. The exception is HADS, where higher scores indicate higher levels of anxiety and/or depression. For each participant, we calculated differentials between post-test and pre-test (Table 2 and Table 3), which were used as dependent variables. The difference between post-test and pre-test (magnitude indicative of intervention-concurrent improvement) was calculated for all instruments except HADS, wherein superior values indicate increased anxiety or depression. Thus, for HADS we did the inverse calculation (pre-post) to homogenize the meaning of differential values.

Given the small sample size and the lack of normal distribution, we used non-parametric analyses. For each standardized test, we compared the pre-post-test differentials across the two groups (guided vs. unguided), with the Mann–Whitney statistical test. The adopted critical level of significance was 0.05. In order to compensate for multiple comparisons, we used Bonferroni corrections.

For the self-report control questionnaire, we categorized participants’ responses and then computed the relative frequency of each category per group. Since this control questionnaire was intended to be merely indicative, no statistical analyses were performed.

## 3. Results

### 3.1. Standardized Tests

Table 3 and Table 4 present the median values of the post-pretest differential for each group. Positive values indicate increased post-intervention functionality, while negative values indicate the opposite. In both groups, there were negative median differentials for MoCA and TMT-AB. In addition, there were negative medians for Stroop tasks in the unguided group (Table 2), and for the quality of life questionnaire (Table 3) in the guided group. 

Cross-group comparisons of post-pre differentials showed that the therapeutic method had significant psychosocial/affective effects in some domains tested by the WHOQOL-BREF (quality of life) questionnaire: psychological domain (*p* = 0.018, *η*2 = 0.471), domain of social relations (*p* = 0.038, *η*2 = 0.414), and general domain 1 (quality of life, *p* = 0.042, *η*2 = 0.406). As shown in Table 3, psychosocial/affective benefits were larger in the unguided group compared to the guided one. Note, however, that these results lose significance when a Bonferroni correction for multiple comparisons is applied. For example, a correction for six comparisons (six questionnaire domains) transforms significance levels into *p* = 0.108 for the psychological domain, *p* = 0.228 for social relations, and *p* = 0.252 for the general domain. A less restrictive correction—considering only the four dimensions relevant to our study (psychological, social, general 1 and general 2) would yield a marginal value (*p* = 0.072) for the psychological domain, and non-significant values (*p* = 0.152, *p* = 0.168) for the remainder.

### 3.2. Self-report Questionnaires

The pattern of responses to the areas targeted by the final control questionnaire is next summarized.

Intervention-unrelated influences: All participants from both groups were engaged in other day-care center activities. An equivalent proportion of patients from each group (two in unguided, five in guided) reported negative life events during the three-month intervention period. Therefore, intervention-unrelated influences seem to have been similar across the two groups, and they are unlikely to account for post-intervention changes. 

Perception of intervention-concurrent changes (Figure 2): Within the psychosocial/affective domain, the proportion of participants reporting improvements in emotional balance and well-being was higher in the unguided group, but participants in the guided group showed more cases of improvement in communication. In the cognitive domain (mental tasks), responses were very similar. Regarding daily activities—which engage the two domains—we saw a larger proportion of improvements in the unguided group.

Perception of artistic outcomes: Two participants out of three in the unguided group said they were satisfied with their art works. In the guided group, seven out of eight expressed satisfaction. When inquired about the meaning of their art works, all participants in both groups referred to calmness and well-being. Two participants in the guided group referred to concentration and attention. 

## 4. Discussion

The purpose of this study was to determine whether guided methods could boost cognitive effects from art therapy and, if so, this would come at the cost of decreasing psychosocial/affective benefits. To that end, we compared the effects of two different art therapy methods—guided vs. unguided. We hypothesized that cognitive benefits would be enhanced by guided methods (hypothesis 1), and psychosocial/affective benefits by unguided (hypothesis 2). Our results provided partial support to hypothesis 2, but not to hypothesis 1: guided methods did not boost cognitive effects, but they seem to have caused some loss in psychosocial/affective benefits.

Regarding hypothesis 1, we saw no cross-group differences for cognitive effects. Responses to the control questionnaire pointed to the same direction, since the proportion of participants reporting perceived enhancements in the ability to execute mental tasks was very similar for the two groups. Regarding hypothesis 2, we saw that the pre-post-test differentials for standardized psychosocial measures were larger in the unguided group than in the guided one. Cross-group differences became marginal after applying corrections for multiple comparisons, but results from the self-report control questionnaire strengthened the idea that the affective/psychosocial domain may benefit from unguided methods: a larger proportion of participants in the unguided group reported increased emotional balance and well-being after therapy, compared to those in the guided group. 

The lack of evidence for increased cognitive benefits in the guided group may indicate that—unlike we hypothesized—externally-provided guidelines are not key to enhancing cognitive functioning. However, we do not think we should rule out this possibility *tout court* because our results may also have due to the particulars of our choices: it is possible that the activities require longer practice times to generate visible effects. It is also possible that, although we attempted to target a specific group of cognitive functions and their corresponding assessment tools (e.g., cognitive inhibition, assessed with the Stroop test), our combination of activities and assessment tools was not optimal. In order to provide answers to this, future studies could manipulate the length of the intervention program, and to consider other assessment tools. 

The tendency for affective/psychosocial benefits being superior under unguided methods (hypothesis 2) may relate to the presence of ideation [37,38,39,40,41]. An additional possibility—that was raised by the comments of one participant in the guided group—is that guided activities induce increased pressure to meet the demands of the task, thus rendering activities more stressful. In order to disentangle these two potential effects—enhancing effect of ideation vs. impairing effect of external guidance—a more complex design than the one we used would be required. Given the exploratory nature of our study, we chose to start small, and thus we used only two groups in our experimental design—unguided, involving ideation and no technical guidance, and guided, involving technical guidance and no ideation. Now that we found effects from this basic manipulation, in the future it would make sense to add at least a third (control) group, focused on a modality of art therapy engaging neither ideation nor external guidance for manual execution. This means that participants would be told what to draw or paint—for instance, a tree or a house (no ideation), but then they would have no guidance on how to do it technically. If these participants showed less psychosocial/affective benefits than unguided ones, this would be evidence for an enhancing effect of ideation. If these benefits were larger than those of guided participants, this would suggest that external guidance can be impairing.

Two exceptions to the superiority of unguided methods concerning psychosocial/affective impact were suggested by questionnaire data: First, a larger proportion of participants in the guided group reported improvements in communication (framed in a psychosocial manner). One explanation may be that the guided group ended up having a larger number of participants, which could have afforded more opportunities within the group. Another explanation may relate to the fact that the guided method, while engaging more external guidance—imposes increased communication between participant and therapist. This is an aspect to consider in future studies, which should balance the amount of communicative exchanges in the two groups. This could be achieved, for instance, by planning and implementing moments of interaction in the unguided group, based on the unfolding creative process. Second, despite showing larger well-being benefits, participants in the unguided group reported less satisfaction with their art works compared to the guided group. On the one hand, this suggests that the feeling of well-being may not arise solely from satisfaction with one’s artistic outputs, which is an interesting idea to explore in future studies. On the other hand, there are some possible explanations for the fact that guided group participants were happier with their work: first, they may have become satisfied for having imitated a famous work of art; second, they may have seen their externally-guided art works as more representative of a fulfilled task, thus embedding increased value. Of course, these differences were small and not supported by statistical tests. They may, however, have heuristic value concerning future research.

From the viewpoint of practical implications, our main findings suggest that unguided methods may be generally advantageous when compared to guided ones: unguided methods seem to have no relative costs, in that they show the relative benefit of increased well-being without causing relative losses in the cognitive domain. In the particular case of cognitively impaired client groups, using guided or unguided methods does not seem to make a difference. Differently, when dealing with psychosocial and affective problems, it seems worthwhile to lend participants freedom to conceive and execute the art piece (unguided method). Of course, this does not mean that ideation coupled with autonomous execution is the only key to well-being, quality of life and positive emotions. For instance, one study with hospital inpatients discharged from intensive care units [65] reported that the act of commenting displayed paintings while adding personal views and personal life recollections reduced anxiety levels among these patients. This suggests that the activation of emotional experiences may play a decisive role in psychosocial/affective recovery regardless of the presence of creation. 

The practical implications of our findings should be faced with caution, since there were several limitations in our study. The first one relates to the very small sample size and the lack of balancing between groups (three vs. eight subjects) that resulted from experimental mortality (drop-out of participants after the study began). Second—and unsurprisingly, given the sample size—our results were not significant when using the most conservative approaches. In view of these limitations, it is possible that affective/psychosocial benefits may be actually larger in the guided group, and that our sample-related limitations did not allow us to see that. This possibility was suggested, for instance, by responses to the communication item of the self-report questionnaire, where guided participants reported increased communication gains. One mechanism subtending this possibility could be, for instance, that external guidance would render the task neither too easy (pressure for fulfilment) nor too hard (no need to take decisions). This is one defining element of the flow state (Chancellor et al., 2014), which could, in this view, have been superior in the guided group. Third, as already pointed out above, we manipulated the therapeutic method, but we manipulated two items simultaneously: the presence vs. absence of ideation and the provision of guidelines for manual execution (or lack of it). As we also referred, this was a conscious start-small choice, associated to the exploratory nature of the current study, but future studies could improve this design by manipulating a single aspect at a time. Finally, our measures were all concentrated at pre-test and post-test moments, following common practice in longitudinal designs. This approach may have overweighed fluctuations in patients’ cognitive functioning occurring at the post-test day and, most of all, fluctuations in their affective states, thus missing the true direction and amount of intervention-concurrent change. In future studies, using more regular measures (e.g., on a weekly basis) could be an option. In such circumstances, the repeated use of standardized tests would not be recommended due to risks of overlearning and manipulating responses, but more informal and/or qualitative measures (e.g., interviews) might be considered.

Our findings contributed to raise new, important questions and research directions. The first one concerns our unresolved riddle: even though guided methods, as implemented here, were unable to outperform unguided ones concerning cognitive enhancement, could it be that longer intervention programs may show an effect? Could it be that other assessment tools, addressing other types of functions, would capture cognitive effects better than the ones we used? Second, what are the sources of the affective/psychosocial benefits that seem to be a privilege of unguided methods: do they arise from ideation, as we first hypothesized, or are they hampered by guidance, as our participants seem to have suggested in their responses to the questionnaire (higher anxiety in the guided group)? Third, our participants were chronic patients with a long disease history (24–29 years). They had considerable autonomy, but they likely underwent several treatments and intervention programs, psychosis was now part of their lives, and thus it is possible that the impact of our program was reduced by these circumstances. Although there are studies showing responsiveness of chronic psychosis patients to art therapy [66], we do not know whether the outcomes of our study would have been different in patients with a shorter illness history. Future studies comparing different illness durations could shed light on this.

Finally, we must not forget that our study was carried out with a sample of patients with schizophrenia, and it is possible that the pattern of therapeutic outcomes we saw does not apply to other clinical conditions such as dementia. For instance, it is possible that—given the type and severity of cognitive symptoms in dementia—guidance may be crucial to allow a minimal level of calmness, and that unguided methods may cause anxiety due to lack of structure. Comparing the effect of guided vs. unguided therapeutic methods in different cognitively impaired populations could, thus, be an important contribution. The question of whether cognitive outcomes may or may not be the same across schizophrenia and dementia fits into the more general question regarding what matters in art therapy intervention research: is it the group (dementia, cancer, etc.) or the domain (cognition, emotion)? So far, research seems to have focused almost exclusively on groups [10]. In the present study, we took the alternative view and focused on domains, using schizophrenia as an example of a clinical condition were both cognition and emotion are affected. Maybe future research could explore further this domain-centered approach, by questioning, for instance, whether the impact of a given art therapy program on cancer-related anxiety is different from the impact on phobia-related anxiety in case the levels of anxiety across these two conditions are the same. 

Outside our main hypotheses, other interesting new questions emerged. One of these concerns the intertwining between communication and guidance: does guidance inevitably increase communication between the members of art therapy sessions? Another concerns the dissociation between personal well-being and satisfaction with own art works: is it real? Is it true that creation may increase well-being regardless of how satisfied one is with the work? Conversely, is it possible that one feels better after imitating than after creating something? 

These questions are sure to open valuable new research avenues for art therapy, which can untangle the mechanisms through which this therapy is effective and thus contribute to better mental health care.

## 5. Conclusions

The present study allowed us to tap into a poorly understood issue in art therapy: how the therapeutic method, namely the amount of externally provided structure, modulates therapy outcomes. We were not able to present evidence in favor of our main hypothesis—that guided methods could boost cognitive effects, but our findings suggested that unguided methods compared to guided ones seem to enhance affective/psychosocial benefits. Therefore, therapeutic methods do seem to matter, and the guided-unguided dichotomy seems to be relevant when it comes to tailoring intervention programs for specific needs. 

## Figures and Tables

**Figure 1 behavsci-10-00065-f001:**
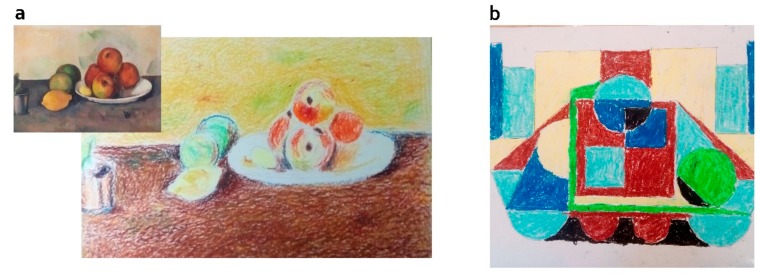
Examples of dry pastel works in the two groups: the guided group (a) recreated one painting by Claude Monet, while the unguided one (b) was given freedom to create using the same technique.

**Figure 2 behavsci-10-00065-f002:**
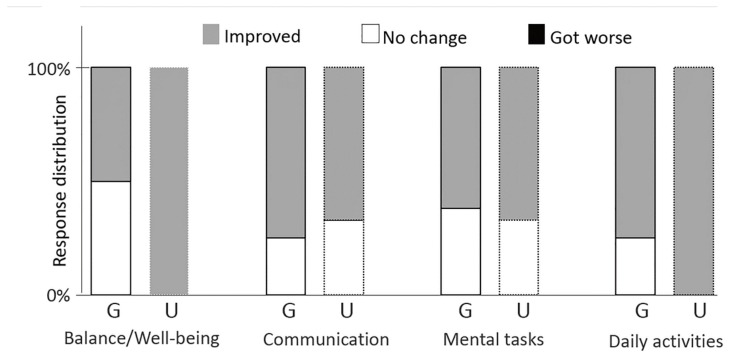
Participants’ perceptions of intervention-concurrent changes as measured by responses to the question “How is your life now as compared to the moment before you started art therapy? Did it improve, got worse, or were there no changes? Consider the domains of psychological stability, communication, ability to perform mental tasks and daily activities.” (G = guided group; U = unguided group).

**Table 1 behavsci-10-00065-t001:** Relevant studies on the impact of art therapy: bold text indicates cognitive target areas.

Reference	Type	Target group	Target areas	Positive outcomes
[14]	Controlled trial	Schizophrenia	Social functioning, interpersonal relations, quality of life, negative symptoms.	Only in negative symptoms
[15]	Controlled trial	Schizophrenia	Self-esteem, participation, agency.	In all areas
[16]	Controlled trial	Schizophrenia and bipolar disorder	Anxiety.	Yes
[17]	Controlled trial	Schizophrenia (inmates)	Anxiety, depression, anger, negative symptoms, compliance with rules and medication, socialization and regular sleeping patterns.	In all areas
[18]	Controlled trial	Schizophrenia	Motivation and pleasure, interpersonal relationships, personal hygiene, hospital program attendance.	In all areas
[19]	Systematic review	Anxious, Depressed, Phobic	Anxiety, depression and quality of life.	In 7 out of 11 studies
[20]	Systematic review	Trauma	Depression and trauma.	In 3 out of 6 studies
[23]	Controlled trial	Diagnosed with breast cancer	Psychological well-being.	Yes
[24]	Systematic review	Healthy or at-risk individuals	Stress.	In 81% of studies
[25]	Controlled trial	Healthy older adults	Negative emotions, self-esteem, anxiety.	In all areas
[26]	Case study	Dementia	Psychological well-being.	Yes
[27]	Controlled trial	Older adults	Depression	Yes
[28]	Controlled trial	Dementia	**Mental acuity**, physical competence, calmness and sociability	In all areas
[29]	Systematic review	Dementia	**Attention**, **memory** and communication.	In 17 studies
[30]	Systematic review	Dementia	Emotional issues and **cognitive decline**.	Only emotional issues
[31]	Systematic review	Dementia	Emotional issues and **cognition**.	In none.

**Table 2 behavsci-10-00065-t002:** Mean (M) and standard deviation (SD) of age, schooling, illness onset and illness duration (in years) in the two groups of participants.

	Guided Group (n = 8)		Unguided Group (n = 3)	
	*M*	*SD*	*M*	*SD*
Age	45.5	5.4	47.6	2.1
Schooling	11.2	1.3	11.3	1.1
Illness onset	20.7	7.1	19.6	6.7
Illness duration	24.7	5.4	29	6.55

**Table 3 behavsci-10-00065-t003:** Comparison of post-pretest differentials between groups for cognitive tests.

	Guided (n = 8)		Unguided (n = 3)			
	Median	Mean rank	Median	Mean rank	*U*	*P*
MoCA	−0.23	6.19	−0.44	5.50	10.5	0.758
Clock-Drawing	0.00	6.38	0.00	5.00	9.00	0.520
Verbal Phonemic Fluency.^a^	0.00	5.63	0.00	7.00	9.00	0.529
Trail-making A	0.33	5.69	0.34	6.83	9.50	0.216
Trail-making B	0.50	6.13	0.00	5.67	11.0	0.836
Trail-making AB^b^	−0.49	6.50	−1.67	4.67	8.00	0.412
Stroop-Word	0.00	6.38	-0.33	5.17	9.50	0.607
Stroop-Colour	0.20	6.69	0.60	4.17	11.0	0.836
Stroop-Colour/Word	0.30	5.94	−0.10	6.17	6.50	0.260
Stroop-Interference	0.10	5.94	−0.20	6.17	11.50	0.919
Spatial localization	0.00	6.38	0.00	5.00	9.00	0.530

a: Verbal / Phonemic Fluency (MoCA), cf. Materials. b: Calculation of the two TMTA / B (A + B) and multiplication (A × B / 100).

**Table 4 behavsci-10-00065-t004:** Comparison of post-pretest differential between groups for the affective/psychosocial domain.

	Guided (n = 8)		Unguided (n = 3)			
	Median	Mean rank	Median	Mean rank		Median
HADS						
Anxiety^a^	−1.00	5.56	0.00	7.17	8.50	0.462
Depression^a^	0.50	5.06	1.00	8.50	4.50	0.115
WHOQOL-BREF						
Physical	−1.78	5.44	8.92	7.50	7.50	0.356
Psychological	−6.25	5.86	8.33	9.83	0.50	**0.018** ^b^
Social relations	−8.33	4.75	16.6	9.33	2.00	**0.038^b^**
Environment	−1.56	6.69	6.25	4.17	6.50	0.259
General domain 1	−20.0	4.88	0.00	9.00	3.00	**0.042^b^**
General domain 2	0.00	6.53	0.00	7.00	9.00	0.498

a: Higher ratings indicate increased anxiety / depression. b: Bold numbers indicate significant differences.

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
