# Peer review of "Exploring the Effects of Guided vs. Unguided Art Therapy Methods"

_behavsci, 2020, doi:10.3390/bs10030065_

Round 1

Reviewer 1 Report

Dear Authors, your manuscript is really interesting and it is about a current topic. I found it clear and complete. I suggest minor language spell check.

Author Response

Dear Authors, your manuscript is really interesting and it is about a current topic. I found it clear and complete. I suggest minor language spell check.

R: Thank you for positive comments. The paper has now been spell-checked by an expert.

Reviewer 2 Report

In the present study the Researchers purpose was to determine if it might possible to maximize the cognitive outcomes of art therapy using guided methods and, in this case, if it's possible to lose or the psychosocial and affective impact of art therapy. To do it, the Authors have tested the hypotheses that (1) guided methods favor cognitive benefits, while (2) unguided therapeutic methods privilege affective and psychosocial ones, implying that the latter will decrease under guided methods.

Overall, I found the paper interesting and timely, but I have some concerns on it that should be addressed prior publication and these are outlined below:

1) I believe that, in the Introduction, the references are a bit outdated. I wonder if there are more recent and rigorous studies on subjects with schizophrenia and bipolar disorders.

2) I suggest to add a Table with more relevant studies on this fiedl in the Introduction.

3) The Authors wrote that "Sixteen participants with a formal diagnosis of schizophrenia and no comorbidities volunteered to take part in this study". But non informations are provided concerning DUP, current treatment, diagnostic methods.

4) How the comorbidities were excluded?

5) The duration of illness is relative long (mean 26y, I presume since forst episode or treatment?). This means that there were chronic subjects and this may impact the results. This point should be discussed.

Author Response

In the present study the Researchers purpose was to determine if it might possible to maximize the cognitive outcomes of art therapy using guided methods and, in this case, if it's possible to lose or the psychosocial and affective impact of art therapy. To do it, the Authors have tested the hypotheses that (1) guided methods favor cognitive benefits, while (2) unguided therapeutic methods privilege affective and psychosocial ones, implying that the latter will decrease under guided methods.

Overall, I found the paper interesting and timely, but I have some concerns on it that should be addressed prior publication and these are outlined below:

  • I believe that, in the Introduction, the references are a bit outdated. I wonder if there are more recent and rigorous studies on subjects with schizophrenia and bipolar disorders.

R: We have now enrichened our text with several new references (highlighted in blue), six of which are post-2017.

  • I suggest to add a Table with more relevant studies on this field in the Introduction.

R: Thank you for your useful suggestion. We added a table (Table 1) with a focus on our main question - the contrast between affective/psychosocial and cognitive outcomes of art therapy.

  • The Authors wrote that "Sixteen participants with a formal diagnosis of schizophrenia and no comorbidities volunteered to take part in this study". But non informations are provided concerning DUP, current treatment, diagnostic methods.
  • How the comorbidities were excluded?

R: We have now specified current pharmacological treatments (lns 190-193), as well as methods of diagnostic and comorbidity exclusion (lns 174-176). We had no direct data on the duration of untreated psychosis (DUP), but we added the age of illness onset (Table 2, lns 193-196), which is quite low (19-20 years) and may suggest that the period of untreated psychosis was short (we assume that initial awareness of the disease was followed by treatment).

5) The duration of illness is relative long (mean 26y, I presume since first episode or treatment?). This means that there were chronic subjects and this may impact the results. This point should be discussed.

R: There is evidence of effectiveness of art therapy in chronic psychiatric patients, as we have now added to the discussion (lns 542-543). Nevertheless, it is true our patients’ illness duration may have affected the results. We have also acknowledged this (lns 539-545), together with a suggestion of future comparative studies (chronic vs. non-chronic patients).

---------------------------------------------------------------------------------

We are grateful to the Reviewer for the sharp comments and useful suggestions.

Reviewer 3 Report

In this study, Costa et al provide interesting insights on the impact of art therapy delivered using two different approaches, guided (with a support) or nonguided. They elegantly show (using a large panel of tests) the lack of differences between the 2 alternates with regard to cognitive impairment/improvement, but a better impact on affective/emotional feelings of the unguided approach. This is a very interesting conclusion.

The study is well designed, clearly introduced, the methods are properley developed and used. Data analysis and treatment is appropriate. The manuscript is well written and easy to read.

This submission may be improved with a lot of minor remarks:

1 The affective/emotional part of art therapy is not only observed with the unguided approach. In a recent study on art therapy consisting in discussion with patients on paintings from a museum, many patients commented displayed paintings while adding personal views and personal life recollections.  This was more pronounced when the discussion focused on self-selected artworks by the patient engaged in the discussion (Int J Envir Res Publ Heatlth 2019, 16, 206; doi:10.3390/ijerph16020206). The authors should add something about this limitation, guided is also impacted by personal emotional factors

2 There are only few patients in each group (8). Thise may probably not allow to drawn definite conclusions.

3 Also, patients are followed up for schizophrenia, and what's apply to this specific disorder may not implicitely apply to other medical conditions such as Alzheimer's disease, depression, pain etc

4 The conclusion section rises a lot of comments. The last words of the first para are not easy to understand, or even confusing ( ..."though this conclusion was limited by the experimental mortality in our study". I believe that this sentence should be removed.

The conclusion should be shortened and probably reduced to the first para only. The hypostheses discussed after should be replaced in the Discussion Section.

Author Response

In this study, Costa et al provide interesting insights on the impact of art therapy delivered using two different approaches, guided (with a support) or nonguided. They elegantly show (using a large panel of tests) the lack of differences between the 2 alternates with regard to cognitive impairment/improvement, but a better impact on affective/emotional feelings of the unguided approach. This is a very interesting conclusion.

The study is well designed, clearly introduced, the methods are properley developed and used. Data analysis and treatment is appropriate. The manuscript is well written and easy to read.

This submission may be improved with a lot of minor remarks:

1 The affective/emotional part of art therapy is not only observed with the unguided approach. In a recent study on art therapy consisting in discussion with patients on paintings from a museum, many patients commented displayed paintings while adding personal views and personal life recollections.  This was more pronounced when the discussion focused on self-selected artworks by the patient engaged in the discussion (Int J Envir Res Publ Heatlth 2019, 16, 206; doi:10.3390/ijerph16020206). The authors should add something about this limitation, guided is also impacted by personal emotional factors

R: Thank you for your suggestion. We added this idea (emotion may be crucial regardless of ideation) to the discussion (lns 500-506).

2 There are only few patients in each group (8). Thise may probably not allow to drawn definite conclusions.

R: Yes. We have now stressed this limitation (lns 507-510).

3 Also, patients are followed up for schizophrenia, and what's apply to this specific disorder may not implicitely apply to other medical conditions such as Alzheimer's disease, depression, pain etc.

R: Yes, that is true. We expanded the discussion on this point and we took the chance to raise the question of targeting groups vs. problem areas, which may be food for thought in future research (lns 546-561).

4 The conclusion section rises a lot of comments. The last words of the first para are not easy to understand, or even confusing ( ..."though this conclusion was limited by the experimental mortality in our study". I believe that this sentence should be removed.

R: We have now removed the sentence.

5 The conclusion should be shortened and probably reduced to the first para only. The hypostheses discussed after should be replaced in the Discussion Section.

R: Yes, that makes sense. We now moved the last two paragraphs of the conclusion into the discussion.

We are grateful to the Reviewer for the sharp comments and useful suggestions.

Round 2

Reviewer 2 Report

Dear Authors, the paper is much improved and worthy of publication.